# Silicon Enhances Plant Resistance of Rice against Submergence Stress

**DOI:** 10.3390/plants10040767

**Published:** 2021-04-14

**Authors:** Taowen Pan, Jian Zhang, Lanmengqi He, Abdul Hafeez, Chuanchuan Ning, Kunzheng Cai

**Affiliations:** 1College of Natural Resources and Environment, South China Agricultural University, Guangzhou 510642, China; pantw@stu.scau.edu.cn (T.P.); zhangjian0167@stu.scau.edu.cn (J.Z.); 201713050105@stu.scau.edu.cn (L.H.); ahafeez1226@scau.edu.cn (A.H.); ningcc@scau.edu.cn (C.N.); 2Guangdong Provincial Key Laboratory of Eco-Circular Agriculture, South China Agricultural University, Guangzhou 510642, China; 3Key Laboratory of Tropical Agro-Environment, Ministry of Agriculture, South China Agricultural University, Guangzhou 510642, China

**Keywords:** rice, submergence, silicon, root morphology, cell structure, antioxidant enzyme

## Abstract

Flooding is an important natural disaster limiting rice production. Silicon (Si) has been shown to have an important role in alleviating varied environmental stress. However, very few studies have investigated the effects and mechanisms of Si in alleviating flood stress in rice. In the present study, wild type rice (cv. Oochikara, WT) and Si-defective mutant (*lsi1*) were chosen to examine the impacts of Si application on plant growth, photosynthesis, cell structure, and antioxidant enzyme activity of rice exposed to submergence stress at tillering stage. Our results showed that Si application improved root morphological traits, and increased Si uptake and plant biomass of WT under submergence stress, but non-significantly influenced *lsi1* mutant. Under submergence stress, leaf photosynthesis of WT was significantly inhibited, and Si application had no significant effects on photosynthetic rate, transpiration rate, stomatal conductance, and intercellular carbon dioxide concentration for both of WT and *lsi1* mutant, but the photochemical quenching of WT was increased and the integrity of cell structure was improved. In addition, Si application significantly reduced malondialdehyde concentration and increased the activity of peroxidase and catalase in WT leaves under submergence stress. These results suggested that Si could increase rice plant resistance against submergence stress by improving root morphological traits and chloroplast ultrastructure and enhancing antioxidant defense.

## 1. Introduction

Rice is an important food crop; over half of the population in the world depend on rice, especially in many Asian and African countries [1]. Global climate change has increased the distribution and frequency of heavy rainfall that can negatively affect plant growth and development. If it persists for a number of days, it may lead to the plant’s death. Many crops, including rice, a semi-aquatic plant, are significantly negatively impacted by flooding, resulting in annual yield loss [2]. More than 20 million hectares of rice in Asia and over 16% of rice in the world are adversely influenced by flooding because of submergence each year [3]. The estimated annual economic loss of this year is more than US$ 600 million. Flooding and reaeration can cause plant oxidative stress, leading to the production and rapid accumulation of reactive oxygen species (ROS) [4]. In addition, the production of ROS may lead to enzyme dysfunction and lipid oxidative damage, and eventually form toxic products, such as malondialdehyde (MDA) [5]. Under environmental stress, plants possess a complex antioxidant defensive system through increasing the activity of antioxidant enzymes such as superoxide dismutase (SOD), peroxidase (POD), and catalase (CAT) to combat harmful effects caused by ROS [4,6,7]. Generally, rice plants adapt to submergence stress through escape or quiescence strategies [8,9]. Under hypoxic conditions, escape strategies for submergence tolerance of rice are usually characterized by rapid internodal or stem elongation and leaf extension, which can be in contact with air [10]. Different environmental and hormonal factors such as ethylene, abscisic acid (ABA), and gibberellic acid (GA) regulate shoot elongation during submergence stress [11]. The rapid elongation of shoot can again restore the contact between leaves and air. However, if the carbohydrate reserves are exhausted during this process, it may cause the plant to die [12]. Specific allelic variants of *Sub1A* confer submergence tolerance in lowland rice, an ethylene-response factor found in some rice varieties that can reduce ethylene and GA responsiveness [13], causing a quiescent growth, which is related to the ability to regenerate after desubmergence [14]. Rice with *Sub1A* adopts a quiescence behavior strategy, where rice plants maintain slow growth and development under flooding conditions, storing energy until the flood water recedes, and then use the stored energy for recovery and extension [15].

Silicon (Si) is the second most abundant element in soil. Rice is a typical Si-accumulating plant, containing large amounts of silicon that are significantly higher than other macronutrients including N, P, and K [16]. Silicon plays an important role in increasing plant resistance against varied biotic and abiotic stresses [17,18]. In the abiotic stress-induced adversity, Si is involved in the regulation of various plant metabolic processes, mainly including the production of osmotic pressure and the regulation of ROS by the antioxidant defense system [19]. Si enhances the resistance of plants to various abiotic stresses by regulating the synthesis/accumulation of plant endogenous hormones. For instance, Si enhanced the activity of polyphenoloxydases, peroxidase, and chitinases in cucumber plant roots infected by *Pythium* [20]. Si deposition in hulls, leaves, and culms enhances the rigidity and strength of a cell wall and reduces transpiration from cuticle, and thus improves the resistance to lodging, UV radiation, and temperature and drought stress [21]. Several researchers have studied the physiological changes and molecular regulation of rice plants exposed to submergence stress [22,23,24]. However, the role and mechanism of Si in alleviating submergence stress is seldom reported. In the present study, we have explored the role of Si in mitigating submergence stress of rice using wild type rice (cv. Oochikara) and Si-defective mutant, which has a mutation in the influx transporter (Lsi1) of Si [25]. Our hypothesis is that Si can alleviate submergence stress of rice through increasing Si uptake and enhancing plant antioxidant system of rice, but will have no effects for Si-defective mutant. The purpose of this study is to investigate the impacts of submergence stress on plant growth as well as cytological and physiological traits of rice, which help decipher the role of Si in mitigating submergence stress.

## 2. Results

### 2.1. Plant Biomass

Plant biomass of wild type (WT) and *lsi1* mutant was significantly reduced under submergence stress (Figure 1). However, exogenous Si application alleviated submergence stress and increased the growth of WT shoot. Under submergence stress, compared with no Si treatment, Si application increased the stem, leaf, and total biomass of WT by 99.8%, 80.6%, and 77.2%, respectively. However, for *lsi1* mutant, Si treatment did not influence root, stem, leaf, and total biomass.

### 2.2. Root Morphological Traits

Submergence stress significantly inhibited root growth for both WT and *lsi1* mutant, while Si application could improve root development and morphological traits of WT (Figure 2). Compared with WT, submergence stress significantly increased the average diameter of *lsi1* mutant. Under submergence stress, surface area and volume of WT were increased by 47.0% and 33.6%, respectively, by adding Si (Table 1). For *lsi1* mutant, Si application had non-significant effects on root morphological traits regardless of submergence treatment.

### 2.3. Silicon Concentration

Si concentrations in stem and leaves of WT were significantly higher than those of *lsi1* mutant regardless of submergence or Si addition (Figure 3). Whatever submergence stress or not, Si application significantly increased Si concentration in the shoots of WT. Si concentrations in the stem and leaves of WT were increased by 27.6% and 39.0%, respectively, under no-submergence stress, and 44.7% and 60.8%, respectively, under submergence stress. However, Si application did not influence Si concentration in stem and leaves of *lsi1* mutant, regardless of submergence stress.

### 2.4. Photosynthesis

Submergence stress significantly reduced *P*_n_, *G*_s_, and *T*_r_ of both WT and *lsi1* mutant, while Si application had non-significant effects (Table 2). In addition, under normal conditions, Si addition significantly reduced the *C*_i_ of WT. However, for *lsi1* mutant, submergence stress and Si application had non-significant effects on *C*_i_.

### 2.5. Chlorophyll Fluorescence

Si application significantly increased *qP* of leaves in WT regardless of submergence stress, but had non-significant effects on *qP* in *lsi1* mutant (Table 3). In addition, submergence stress significantly increased *qN* of WT, while it reduced Φ_PSII_. Si application and submergence stress did not influence *F*_o_, *F*_m_, and *F*_v_/*F*_m_ of both rice materials.

### 2.6. Chloroplast Ultrastructure

Ultrastructural changes in the chloroplast were observed in WT and *lsi1* mutant (Figure 4). Transmission electron microscope (TEM) observation revealed that, under normal growth conditions, the ultrastructure of WT and *lsi1* mutant was complete and clear. In addition, the proportion of chloroplasts was large, which were fusiform in shape. The thylakoids were rich, and the lamella structure was well developed and neatly arranged (Figure 4A,D). Compared with the normal condition (CK), significant changes in the ultrastructure of rice leaves were presented under submergence treatment (Figure 4B,E). Under submergence stress, there were a large number of starch grains in WT chloroplast, and partial separation between chloroplasts and cell walls (Figure 4B). The chloroplast of *lsi1* mutant showed a hemispherical shape and tended to decompose. In addition, the chloroplast was completely separated from the cell wall, and the number of osmophilic particles increased greatly (Figure 4E). Si application gradually restored the appearance and structure of WT chloroplast (Figure 4C). In addition, Si application could also reduce the number of starch grains in chloroplast. However, Si application had non-significant effects on the appearance and structure of *lsi1* mutant (Figure 4F).

### 2.7. Malondialdehyde Concentration and Antioxidant Enzyme Activity

When exposed to submergence stress, MDA concentrations in the leaves of WT and *lsi1* mutant were significantly increased (Table 4). Si application significantly reduced MDA concentration of WT regardless of stress, but had a non-significant effect on MDA concentration of *lsi1* mutant. Compared with WT, submergence stress significantly increased POD activity of *lsi1* mutant, but had a non-significant effect on SOD and CAT activity. Si application and submergence stress had non-significant effects on SOD activity in the leaves of WT and *lsi1* mutant. In addition, Si application did not influence POD and CAT activity in non-submergence stressed leaves of WT and *lsi1* mutant. However, under submergence stress, Si application increased POD activity and CAT activity in WT by 41.0% and 40.8%, respectively, but had a non-significant effect on *lsi1* mutant.

## 3. Discussion

### 3.1. Si Application Increased Si Uptake and Improved Plant Growth under Submergence Stress

The submergence tolerance of rice is a complex trait influenced by the interaction between environmental conditions and rice genotypes [9,26]. Our results showed that submergence stress not only inhibited root growth of both WT and *lsi1* mutant (Figure 2), but also reduced biomass accumulation (Figure 1). Roots play an important role in nutrient absorption under submergence stress, and are also one of the carbohydrate storage functions required for plant survival [27]. In the present study, increased average diameter may be related to the aerenchyma tissue formed in the root.

Numerous studies have reported that Si uptake ability of rice roots contributes to Si accumulation in rice shoot [25]. The present study found that Si application significantly increased shoot Si concentration and uptake of Si regardless of stress for WT, not *lsi1* mutant (Figure 3), which is beneficial to plant resistance. However, submergence stress had a non-significant effect on Si uptake in WT and *lsi1* mutant. *Lsi1* is an encoded transporter mainly expressed in roots and has specific transport activity for silicon; it is located in the plasma membrane of cells in both the exodermis and endodermis. In addition, as an influx transporter, *Lsi1* is responsible for transporting silicon from external solutions to root cells [28]. Ma et al. [29] reported that Si uptake of Si-defective mutant was significantly reduced compared with vector control plants. Therefore, knocking out the transporter gene will cause loss of Si absorption [29]. Our results showed that Si absorption of *lsi1* mutant was significantly lower than that of WT, regardless of submergence or Si addition (Figure 3), which is similar to the results by Ma et al. [29], who reported that *lsi1* mutant accumulated less silicon in the shoot during the growth period. Under submergence stress, plants form aerenchyma tissue in roots and leaves, which promotes the exchange of gases underwater. This not only helps to transport O_2_ to the root system, but also exhausts the waste gases produced by the root system, such as methane, hydrogen sulfide, carbon dioxide, and so on. We found that Si application significantly increased plant biomass accumulation of WT (Figure 1), similar to the findings of Chu et al. [30] and Viciedo et al. [31]. Chu et al. [30] demonstrated that, under submergence conditions, basal application of Si significantly reduce the loss of rice biomass. Viciedo et al. [31] reported that, under ammonium toxicity, adding Si to the nutrient solution alleviated the growth reduction of radish seedlings.

In addition, our study demonstrated that Si treatment significantly increased the surface area and volume of the WT root system, which was similar to the results by Fan et al. [32] and Li et al. [33], who found that Si improved root morphological traits of rice and tomato under heavy metal and salinity stress, respectively. It is well known that Si acts as a mechanical barrier to promote root elongation and protect the stele by hardening the cell wall of endodermal tissues and stele [34,35]. Biju et al. [36] reported that the application of Si increased the Si concentration in lentil under drought stress, which may be caused by the deposition of Si in the cell wall. The deposited Si can strengthen the membrane of plant cells and change its permeability, thereby improving drought resistance.

### 3.2. Si Application Did Not Influence Leaf Photosynthesis under Submergence Stress

The leaf gas exchange of plants is extremely sensitive to submergence stress [37]. Under submergence stress, the photosynthetic capacity of leaf is also limited owing to the shading effect of water [38]. Our study showed that submergence stress negatively affected the photosynthesis in WT and *lsi1* mutant (Table 2). In addition, several studies reported that Si application improved plant photosynthesis under drought and heavy metal stress [39,40,41]. However, in the present study, Si application had non-significant effects on leaf photosynthesis of WT under submergence stress (Table 2).

Chlorophyll fluorescence plays an important role in photosystem II (PSII) activity and changes in photosynthetic metabolism of plants under stress [42]. In addition, PSII activity is highly sensitive to floods and water-logging [37]. In the present study, submergence stress showed significant effects on the PSII activity in rice, as reflected by the decreased *qP* and Φ_PSII_, and increased *qN* (Table 3), indicating that submergence stress inhibited photosynthetic electron transfer, resulting in a reduction of photosynthesis, which is consistent with the results of flooding stress change of *Populus simonii* [43]. Studies found that Si application could improve chlorophyll fluorescence of maize plants under drought stress [44] and tomato plants under salt stress [45]. In addition, Nwugo and Huerta [46] found that the addition of Si induced a significant increase in *qP* of rice plants under Cd stress. It was also observed that Si application significantly increased *qP* in WT. Kaufman et al. [47] proposed the “window hypothesis” of Si, and believed that Si deposited in the leaf epidermal cells in the form of Si bodies could be used as a “window” to increase light-use-efficiency through accelerating light transmission to the photosynthetic mesophyll tissues. However, our results showed that Si application had a non-significant effect on Φ_PSII_ of WT and *lsi1* mutant (Table 3), and the reasons need to be further studied.

### 3.3. Si Addition Improved Leaf Chloroplast Structure under Submergence Stress

In higher plants and algae, the chloroplast contains a highly ordered thylakoid membrane system that provides structural properties for optimal light capture [48]. In addition, Bertamini et al. [49] reported that the structural integrity of thylakoids contributes to the conversion of light energy in chloroplast photosynthesis. In the present study, submergence stress resulted in an increase of osmophilic granules in *lsi1* mutant. Si application had non-significant effect on the distribution of osmophilic granules in *lsi1* mutant (Figure 4F), while reducing the number of starch grains in chloroplasts of WT and the destruction of thylakoids (Figure 4C), which helped to restore the structural integrity of chloroplasts and improve photosynthesis. Mittelheuser and Van Steveninck [50] found that, when the chloroplast membrane structure was well developed, osmium particles were absent or rarely present. In addition, Guo et al. [51] found that Si application was helpful for the recovery of chloroplast appearance and structure of rice seedlings under Cd stress. The quantities and sizes of starch grains and osmiophilic granules in the cells were radically reduced by Si [51]. Song et al. [41] and Li et al. [52] also found that Zn and Mn destroyed the ultrastructure of chloroplasts, while the addition of Si alleviated the hazardous effects of heavy metals on rice. All in all, a complete and orderly cell structure is the basis of sustaining normal functions of cells, providing appropriate conditions for different molecules and enzymes [51].

### 3.4. Si Addition Reduced Oxidative Damage and Enhanced the Activity of POD and CAT under Submergence Stress

When plants are under stress, the antioxidant system will produce a series of stress responses, leading to changes in the activity of antioxidant enzymes [5]. In addition, in the antioxidant enzyme system, POD and CAT have a certain ability to decompose H_2_O_2_ produced under stress, and achieve the purpose of eliminating active oxygen in plants [53,54]. Our results showed that submergence stress caused oxidative damage for both materials; the concentration of MDA was increased especially for *lsi1* mutant (Table 4). Huang et al. [55] also found that MDA concentration in rice leaves increased under Cd and Zn stress. Numerous studies have shown that, under stress conditions, Si application significantly reduced MDA concentration in plant organs and alleviated the damage to plants [56,57,58,59,60,61]. Our results were consistent with those studies (Table 4), indicating that Si treatment decreased membrane lipid peroxidation in rice. In addition, submergence stress significantly increased POD activity of WT leaves, and Si application could enhance its activity (Table 4). However, the CAT activity of WT was significantly reduced under submergence stress. This is because, when the submergence stress exceeded a certain intensity, the rice cannot scavenge oxygen free radicals in time and the antioxidant capacity of antioxidant enzymes was limited, resulting in decreased enzyme activity. However, under submergence stress, the activity of CAT in *lsi1* mutant was not significantly reduced, possibly because non-enzymatic antioxidants such as glutathione (GSH) and ascorbic acid (ASA) were involved in the removal of H_2_O_2_ to cope up with submergence stress. Under different stresses and intensities, the changes of antioxidant enzymes in plants are different. For example, Ma et al. [39] reported that drought stress resulted in the increase of POD activity of cucumber plants, while silicon-treated plants showed lower POD activity than that of no-silicon-treated plants. In addition, CAT activity in cucumber plants was decreased significantly with the increase of drought intensity, and CAT activity in silicon-treated plants was higher than that of no-silicon-treated plants. However, Huang et al. [55] observed that SOD activity decreased in rice leaves, while POD and CAT activity increased under heavy metals stress. In different growth stages, the addition of Si reduced the intensity of these changes caused by Cd and Zn stress. These results show that Si takes an important part in improving the self-protection mechanism of rice against submergence stress through the antioxidant enzyme system.

## 4. Materials and Methods

### 4.1. Planting Material and Growth Conditions

Pot experiment was conducted at the Ecological Farm of South China Agricultural University, Guangzhou, China (113°21′ E, 23°09′ N). Wild type rice (cv. Oochikara, WT) and Si-defective mutant rice (*lsi1*) were used in the study. This mutant was defective in active Si uptake. So, this rice was defined as a Si-defective mutant. Seeds were soaked for 30 min in 10% hydrogen peroxide, rinsed three times with deionized water, and then placed in incubator for 2 days at 80% humidity and 25℃ temperature. After germination, seedlings were transferred to a paddy field for growth. Plastic pots (upper diameter 27.5 × lower diameter 17.5 × height 20 cm) were filled with 5 kg of soil. Each pot was planted with three holes and two seedlings (fourth-leaf stage) planted in each hole. The basic properties of soil for pot experiments were as follows: pH 5.72, organic matter 18.36 g kg^−1^, total N 1.16 g kg^−1^, total P 0.61 g kg^−1^, total K 9.72 g kg^−1^, and available Si 147.28 mg kg^−1^.

### 4.2. Experimental Design

The experiment included four treatments with three biological replicates in the study: (1) no submergence without silicon application (CK); (2) no submergence with silicon application (Si); (3) submergence without silicon application (Sub); and (4) submergence with silicon application (Si+Sub). Silicon (K_2_SiO_3_) was added before transplanting, and the concentration was 2 mmol kg^−1^. The effect of potassium was eliminated by adding the same amount of KCl_3_. In the present experiment, to fulfill nutrient requirements for normal rice growth, nitrogen (CH_4_N_2_O; 0.325 g kg^−1^), phosphorus (CaP_2_H_4_O_8_; 0.243 g kg^−1^), and potassium (KCl_3_; 0.241 g kg^−1^) were added and mixed in the soil thoroughly for all treatments. For those submerged treatment, rice plants, 14 days after transplantation, along with pot were placed in a bucket (upper diameter 50 × lower diameter 40 × height 62 cm) for 7 days; the submerged depth was two-thirds of the average plant height. After the submerged treatment, plant samples were collected to analyze biomass, root traits, Si content, photosynthesis traits, chlorophyll fluorescence, chloroplast ultrastructure, MDA concentration, and antioxidant enzyme activity.

### 4.3. Determination of Biomass and Root Morphological Traits

Sampled rice plants were separated into root, stem, and leaf; initially dried at 110 °C for 30 min; and then at 65 °C till the samples attained a constant weight. Root, stem, and leaf biomass was determined using electric balance. Fresh root samples were collected to scan (using scanner Epson Expression 1600 pro, Model EU-35, Suwa, Japan), then root morphological traits including root surface area, length, diameter, and volume were analyzed by WinRHIZO Reg. 2009 (Regent Instruments, Inc., Quebec, QC, Canada).

### 4.4. Measurement of Si Concentration

Colorimetric molybdenum blue method was used to determine Si concentration in rice stem and leaf [62]. In short, 0.3 g samples of stems and leaves in rice were ashed at 550 °C for 3 h, then the ash was dissolved using 1.3% hydrogen fluoride, and Si concentration was determined by spectrophotometer (PGENERAL TU-1901 UV-VIS, Beijing, China) at 811 nm.

### 4.5. Determination of Photosynthesis

LI-6400XT photosynthetic system (LI-COR, Lincoln, NE, USA) with a photon flux density of 1000 μmol m^−2^ s^−1^ and a flow rate of 500 μmol s^−1^ was used to measure the photosynthetic index of the third fully expanded leaf, including the photosynthetic rate (*P*_n_), transpiration rate (*T*_r_), intercellular carbon dioxide concentration (*C*_i_), and stomatal conductance (*G*_s_).

### 4.6. Chlorophyll Fluorescence Measurements

Chlorophyll fluorometer (MINI-PAM-II, Walz, Germany) was used to measure the chlorophyll fluorescence of the third fully expanded leaf. The actinic light intensity and the saturated flash intensity were set to 190 µmol m^−2^ s^−1^ and 6000 µmol m^−2^ s^−1^, respectively. The light induction curve program was used to measure on the plants after 20 min of dark treatment. Chlorophyll fluorescence parameters include *F*_o_ (minimal fluorescence), *F*_m_ (maximal fluorescence), Φ_PSII_ (actual photochemical efficiency of PSII), *F*_v_/*F*_m_ (maximal photochemical efficiency), *qP* (photochemical quenching), and *qN* (non-photochemical quenching). *F*_v_/*F*_m_, Φ_PSII_, *qP*, and *qN* are calculated as follows: *F*_v_/*F*_m_ = (*F*_m_ − *F*_o_)/*F*_m_; Φ_PSII_ = Δ*F*/*F*_m_′ = (*F*_m_′ − *F*)/*F*_m_′; *qP* = (*F*_m_′ − *F*)/(*F*_m_′ − *F*_o_′); *qN* = 1 − (*F*_m_′ − *F*_o_′)/(*F*_m_ − *F*_o_).

### 4.7. Microscopic Observation of Chloroplast Structure

Fresh leaves were collected and cut into pieces (0.5 × 0.3 cm). The pieces were fixed with 5% glutaraldehyde at 4 °C for 8 h and rinsed four times with phosphate buffer (0.1 M, pH 7.2) for 15 min each time. Then, they were fixed overnight with 1% OsO_4_ and rinsed four times with phosphate buffer solution (PPB) for 15 min each time. After the gradient concentration (30%, 50%, 70%, 80%, 90%, and 100%), the ethanol solution was dehydrated for 15 min, and then dehydrated with 100% ethanol solution for another 15 min. The material was treated with 100% acetone twice for 15 min each time and embedded with different ratios of acetone and resin, and polymerized at 70 °C for 24 h. Finally, the embedded material was cut with a microtome (Leica, Germany) and observed using a transmission electron microscope (TEM) (Talos L120C, Waltham, MA, USA).

### 4.8. Determination of Malondialdehyde Concentration

MDA concentration in leaves was determined by thiobarbituric acid (TBA) reaction [63]. Here, 0.5 g leaf sample was extracted with 10% (*w*/*v*) trichloroacetic acid (TCA), and 0.5% TBA was added to the extract. After 20 min of boiling water bath, centrifugation was carried out. The absorbance of the sample supernatant was measured at 450 nm, 532 nm, and 600 nm at the same time.

### 4.9. Antioxidant Enzyme Activity Measurements

SOD activity in leaves was measured according to its ability to inhibit photochemical reduction of Nitrotetrazolium Blue chloride (NBT) [64] at 560 nm using a spectrophotometer (PGENERAL TU-1901 UV-VIS, Beijing, China). POD activity was determined by the Guaiacol method of Egley et al. [65]. The increase in absorbance of the reaction system in 5 min was measured at an absorbance of 470 nm. CAT activity was determined by the decomposition rate of H_2_O_2_ at 240 nm [66].

### 4.10. Statistical Analysis

All data were expressed as the mean ± standard error for three replicates. Data were analyzed using SPSS 18.0 software (SPSS Inc., Chicago, IL, USA), and data differences among the treatments were evaluated by one-way analysis of variance (ANOVA) and Duncan’s multiple range test (MRT) at a 0.05 probability level.

## 5. Conclusions

In summary, our results suggested that submergence stress had negative impacts on the growth and development of rice at tillering stage. Si application could reverse the inhibited effect of submergence stress through increasing Si uptake and accumulation and plant biomass, improving root morphological traits and chloroplast ultrastructure. In addition, Si reduced oxidase damage by enhancing the activity of POD and CAT and reducing MDA concentration, thereby alleviating the damage of submergence stress to rice. Further studies are needed to decipher the molecular mechanisms of Si-alleviated submergence stress.

## Figures and Tables

**Figure 1 plants-10-00767-f001:**
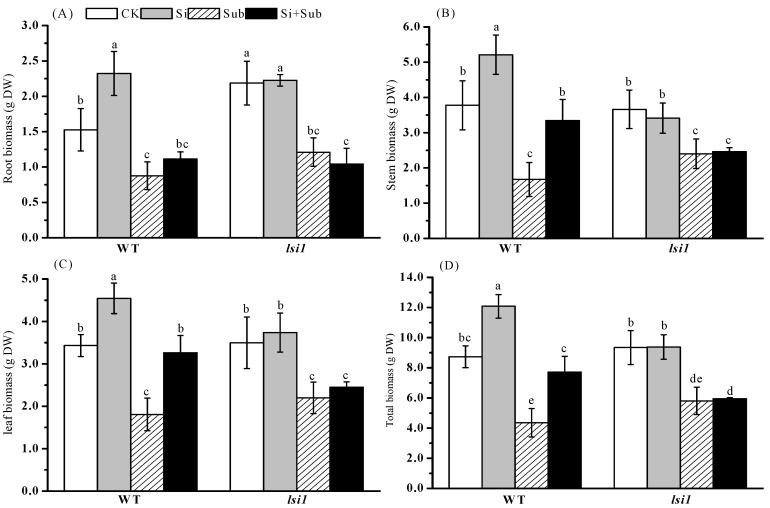
Effects of Si application on root (**A**), stem (**B**), leaf (**C**), and total biomass (**D**) of rice under submergence stress. Values are expressed as mean ± SE from three replicates (n = 3). The different letters above the bars indicate significant differences among all treatments (*p* < 0.05). WT, wild type; DW, dry weight.

**Figure 2 plants-10-00767-f002:**
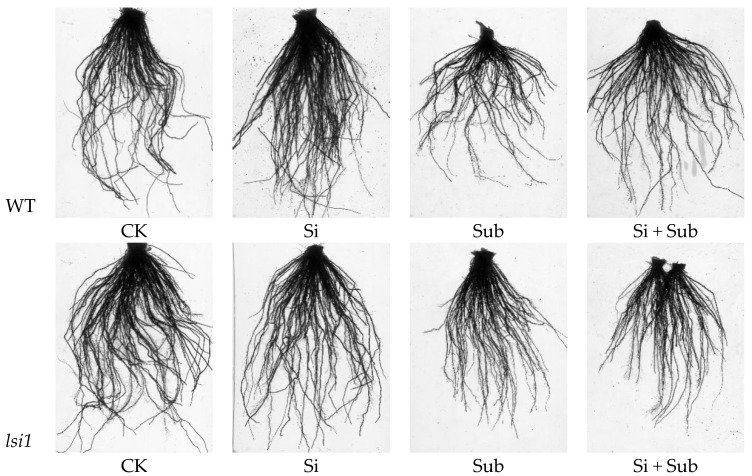
Effects of Si application on phenotypic characteristics of roots under submergence stress. WT, wild type.

**Figure 3 plants-10-00767-f003:**
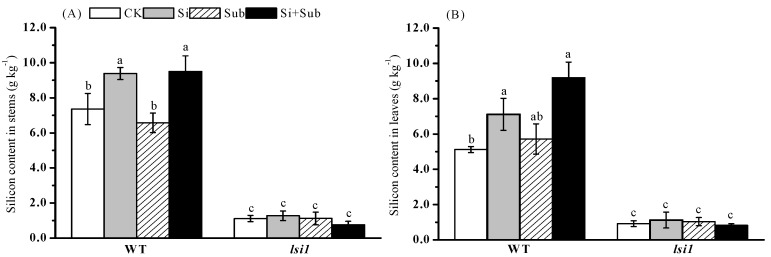
Effects of Si application on Si concentration in rice stem (**A**) and leaf (**B**) under submergence stress. Values are expressed as mean ± SE from three replicates (n = 3). The different letters above the bars indicate significant differences among all treatments (*p* < 0.05). WT, wild type.

**Figure 4 plants-10-00767-f004:**
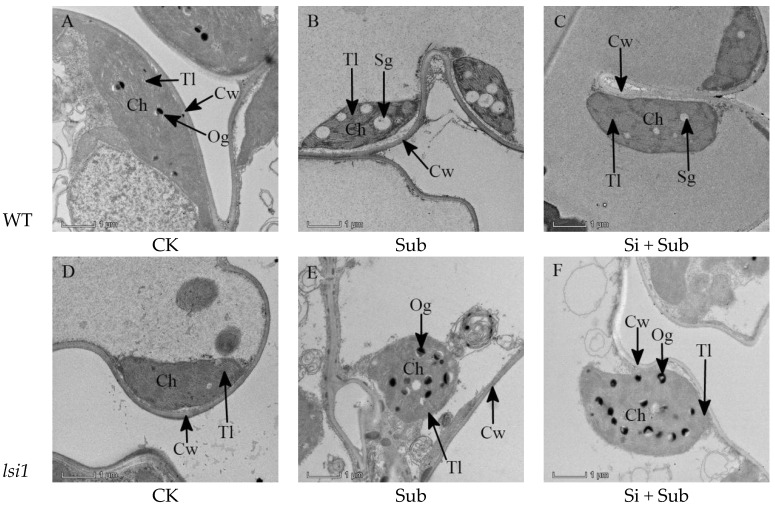
Effects of Si application on chloroplast ultrastructure in leaves under submergence stress (×8500, bar = 1 μm). WT, wild type. (**A**–**C**), WT; (**D**–**F**), *lsi1*; (**A**,**D**), CK; (**B**,**E**), Sub; (**C**,**F**), Si + Sub. Ch, chloroplast; Cw, well wall; Tl, thylakoild lamella; Og, osmiophilic granules; Sg, starch grains.

**Table 1 plants-10-00767-t001:** Effects of Si application on root morphological traits of rice under submergence stress.

Materials	Treatment	Total Root Length (m)	Surface Area (cm^2^)	Volume (cm^3^)	Average Diameter (mm)
WT	CK	123.5 ± 12.0 ^b^	1150.5 ± 157.4 ^bc^	9.03 ± 1.61 ^b^	0.33 ± 0.01 ^cd^
	Si	175.8 ± 16.8 ^a^	1529.2 ± 228.0 ^a^	11.16 ± 0.15 ^a^	0.32 ± 0.01 ^cde^
	Sub	57.0 ± 1.7 ^cd^	622.8 ± 75.1 ^d^	5.65 ± 0.37 ^e^	0.35 ± 0.02 ^bc^
	Si + Sub	77.9 ± 6.1 ^c^	915.7 ± 56.3 ^c^	7.55 ± 0.89 ^bcd^	0.37 ± 0.01 ^b^
*lsi1*	CK	118.2 ± 25.1 ^b^	1125.8 ± 198.7 ^bc^	8.85 ± 1.24 ^bc^	0.32 ± 0.01 ^de^
	Si	140.4 ± 21.3 ^b^	1247.6 ± 98.8 ^b^	9.15 ± 0.68 ^b^	0.30 ± 0.02 ^e^
	Sub	49.7 ± 7.0 ^d^	621.3 ± 104.6 ^d^	7.18 ± 0.55 ^cde^	0.41 ± 0.01 ^a^
	Si + Sub	46.8 ± 7.8 ^d^	516.9 ± 53.4 ^d^	6.01 ± 0.98 ^de^	0.42 ± 0.02 ^a^

Values are expressed as mean ± SE from three replicates (n = 3). The different letters in the same column indicate significant differences among all treatments (*p* < 0.05). WT, wild type.

**Table 2 plants-10-00767-t002:** Effects of Si application on photosynthesis of rice leaves under submergence stress.

Materials	Treatment	*P*_n_ (μmol CO_2_ m^−^^2^ s^−^^1^)	*G*_s_ (μmol H_2_O m^−^^2^ s^−^^1^)	*C*_i_ (μmol CO_2_ mol^−^^1^)	*T*_r_ (mmol H_2_O m^−^^2^ s^−^^1^)
WT	CK	15.49 ± 0.62 ^a^	0.53 ± 0.08 ^a^	338.24 ± 8.83 ^a^	6.23 ± 0.63 ^de^
	Si	15.03 ± 0.32 ^a^	0.44 ± 0.02 ^ab^	317.35 ± 3.84 ^b^	7.35 ± 0.34 ^bcd^
	Sub	12.06 ± 0.40 ^bc^	0.28 ± 0.04 ^d^	302.15 ± 9.25 ^bc^	6.33 ± 0.60 ^de^
	Si + Sub	11.53 ± 1.22 ^c^	0.24 ± 0.05 ^d^	295.41 ± 9.20 ^c^	6.04 ± 0.82 ^e^
*lsi1*	CK	13.33 ± 0.73 ^b^	0.40 ± 0.09 ^bc^	316.52 ± 10.96 ^b^	8.54 ± 0.43 ^a^
	Si	12.75 ± 1.79 ^bc^	0.30 ± 0.02 ^cd^	301.33 ± 10.83 ^bc^	8.44 ± 0.69 ^ab^
	Sub	11.06 ± 0.45 ^c^	0.30 ± 0.02 ^cd^	311.63 ± 8.84 ^bc^	7.83 ± 0.56 ^abc^
	Si + Sub	11.55 ± 0.59 ^c^	0.25 ± 0.06 ^d^	293.43 ± 15.75 ^c^	6.95 ± 0.64 ^cde^

Values are expressed as mean ± SE from three replicates (n = 3). The different letters in the same column indicate significant differences among all treatments (*p* < 0.05). WT, wild type. *P*_n_, photosynthetic rate; *T*_r_, transpiration rate; *C*_i_, intercellular carbon dioxide concentration; *G*_s_, stomatal conductance.

**Table 3 plants-10-00767-t003:** Effects of Si application on chlorophyll fluorescence of rice under submergence stress.

Materials	Treatment	*qP*	*qN*	*F* _o_	*F* _m_	*F*_v_/*F*_m_	Φ_PSII_
WT	CK	0.533 ± 0.009 ^bc^	0.628 ± 0.050 ^c^	511.7 ± 34.6 ^a^	3363.3 ± 202.1 ^a^	0.848 ± 0.003 ^a^	0.379 ± 0.005 ^a^
	Si	0.559 ± 0.012 ^a^	0.672 ± 0.027 ^bc^	501.7 ± 14.0 ^a^	3223.7 ± 28.9 ^a^	0.844 ± 0.004 ^a^	0.383 ± 0.004 ^a^
	Sub	0.529 ± 0.008 ^c^	0.737 ± 0.023 ^a^	524.3 ± 14.0 ^a^	3394.3 ± 117.7 ^a^	0.845 ± 0.007 ^a^	0.345 ± 0.018 ^b^
	Si + Sub	0.551 ± 0.014 ^ab^	0.731 ± 0.017 ^ab^	515.7 ± 18.6 ^a^	3261.3 ± 187.3 ^a^	0.842 ± 0.008 ^a^	0.357 ± 0.002 ^ab^
*lsi1*	CK	0.556 ± 0.010 ^a^	0.742 ± 0.039 ^a^	496.0 ± 19.0 ^a^	3148.3 ± 66.9 ^a^	0.842 ± 0.003 ^a^	0.357 ± 0.021 ^ab^
	Si	0.536 ± 0.016 ^bc^	0.750 ± 0.040 ^a^	494.3 ± 28.7 ^a^	3159.0 ± 127.0 ^a^	0.844 ± 0.004 ^a^	0.342 ± 0.017 ^b^
	Sub	0.533 ± 0.005 ^bc^	0.733 ± 0.019 ^ab^	497.3 ± 12.6 ^a^	3226.3 ± 102.6 ^a^	0.846 ± 0.003 ^a^	0.348 ± 0.010 ^b^
	Si + Sub	0.521 ± 0.010 ^c^	0.722 ± 0.036 ^ab^	498.0 ± 5.2 ^a^	3234.7 ± 96.2 ^a^	0.846 ± 0.005 ^a^	0.344 ± 0.021 ^b^

Values are expressed as mean ± SE from three replicates (n = 3). The different letters in the same column indicate significant differences among all treatments (*p* < 0.05). WT, wild type. *qP*, photochemical quenching; *qN*, non-photochemical quenching; *F*_o_, minimal fluorescence; *F*_m_, maximal fluorescence; *F*_v_/*F*_m_, maximal photochemical efficiency; Φ_PSII_, actual photochemical efficiency of PSII.

**Table 4 plants-10-00767-t004:** Effects of Si application on malondialdehyde (MDA) concentration and antioxidant enzyme activity of rice under submergence stress.

Materials	Treatment	MDA Concentration (nmol g^−^^1^ FW)	SOD Activity (U g^−^^1^ FW)	POD Activity (U min^−^^1^ g^−^^1^ FW)	CAT Activity (U min^−^^1^ g^−^^1^ FW)
WT	CK	20.0 ± 1.3 ^d^	22.2 ± 0.2 ^a^	999.3 ± 176.5 ^c^	1502.2 ± 233.3 ^ab^
	Si	15.1 ± 1.1 ^e^	22.1 ± 0.7 ^a^	1129.6 ± 246.1 ^c^	1751.1 ± 188.7 ^a^
	Sub	27.8 ± 1.4 ^b^	20.2 ± 2.3 ^a^	1638.5 ± 231.3 ^b^	1017.8 ± 244.4 ^d^
	Si + Sub	24.6 ± 1.2 ^c^	20.7 ± 2.0 ^a^	2311.0 ± 406.7 ^a^	1432.9 ± 60.0 ^abc^
*lsi1*	CK	17.6 ± 0.9 ^d^	22.2 ± 0.4 ^a^	938.5 ± 147.6 ^c^	1126.7 ± 54.6 ^cd^
	Si	19.4 ± 1.2 ^d^	22.6 ± 0.3 ^a^	1046.7 ± 72.8 ^c^	1047.8 ± 180.3 ^d^
	Sub	32.2 ± 2.5 ^a^	20.8 ± 1.9 ^a^	2417.8 ± 220.3 ^a^	1048.9 ± 274.0 ^d^
	Si + Sub	32.7 ± 0.8 ^a^	22.1 ± 0.2 ^a^	2434.1 ± 76.3 ^a^	1180.0 ± 30.6 ^bcd^

Values are expressed as mean ± SE from three replicates (n = 3). The different letters in the same column indicate significant differences among all treatments (*p* < 0.05). WT, wild type. SOD, superoxide dismutase; POD, peroxidase; CAT, catalase. FW, fresh weight.

## Data Availability

No new data were created or analyzed in this study. Data sharing is not applicable to this article.

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
