# Peer review of "Silicon Enhances Plant Resistance of Rice against Submergence Stress"

_plants, 2021, doi:10.3390/plants10040767_

Round 1
Reviewer 1 Report
In this study, the author described that silicon enhances the plant resistance of rice against submergence stress. In this study, wild-type rice (cv. Oochikara, WT) and Si-detective mutant (lsi1) were chosen to examine the impacts of Si application on plant growth, photosynthesis, cell structure, and antioxidant enzyme activity of rice exposed to submergence stress at tillering stage. Their results disclosed that Si application improved root morphological traits, increased Si content, and plant biomass of WT under submergence stress but non-significantly influenced lsi1 mutant. Under submergence stress, Si application had non-significant effects on photosynthetic rate (Pn), transpiration rate (Tr), stomatal conductance (Gs), and intercellular carbon dioxide concentration (Ci) for both of WT and lsi1 mutant, but the photochemical quenching (qP) of WT was increased, and the integrity of cell structure was improved. In addition, Si application significantly reduced malondialdehyde (MDA) concentration, increased the peroxidase (POD) activity, and catalase (CAT) activity in WT leaves under submergence stress. Manuscript have many grammatical errors.
The manuscript does not explain about lsi1 mutant where it has insertion, how authors have characterized this mutant homozygosity, and expression results. To be frank, one mutant is not enough to describe the function of the gene. The authors need to check the allelic mutants or RNAi, or Ox lines.
L38 resulting in to annual yield loss to resulting in annual yield loss.
L64 significant higher than to significantly higher than.
L65 Silicon plays important role in increasing to Silicon plays an important role in increasing.
L71 enhances the rigidity to enhance the rigidity.
L17, L77 rice wild type to wild type rice.
L81 were significantly reduced to was significantly reduced.
L211 but had non-significant effect to but had a non-significant effect.
L266 Chlorophyll fluorescence play to Chlorophyll fluorescence plays.
L334 A wild type rice to Wild type rice.
L379 flash intensity were set to flash intensity was set.
L416 had a negative to had negative.
I found plagiarism ( 14% to author previous article “Silicon Amendment Reduces Soil Cd Availability and Cd Uptake of Two Pennisetum Species “) at L22-23, L92-93, L133-136, L145-146, L163-164, L215-216, L374-375, L401-402, L411-414.
Reviewer 2 Report
Silicon protects plants from abiotic stresses such as high temperatures, drought and lodging, and biotic stresses such as pathogens. However, the role of silicon in rice under submergence condition has not been investigated in detail. The authors used WT and lsi1 mutant to clarify the importance of silicon during submergence. The authors showed that silicon is involved in root growth, improvement of chloroplast structure, and reduction of oxidative stress under submergence by physiological, morphological and biochemical approaches. Although the authors have not clarified the molecular mechanism of how silicon sustains plant growth during submergence, this study provides new insights into the importance of silicon in rice during submergence, in addition to the previously reported importance in biotic and abiotic stress tolerances. I have some comments which may improve the manuscript.
- Although it is described as SE in the Figure legend, it is described as standard deviation in the chapter on Materials and methods. Which one is actually shown in the figure? Please clarify this point. The experiment seems to be done only with n=3. Are replicates defined as biological or technical?
- It is better to describe what Pn, Gs, etc. mean in the caption of Table 2. This is applicable to Table 3.
Reviewer 3 Report
Very few studies examine the effect of silicone in ameliorating flooding stress response in plants. In this respect the manuscript contains some new information. However, major revision is necessary in order to be published, especially in research justification, data interpretation and search for underlying mechanisms.
The Introduction should be more informative concerning the mechanism of the beneficial effects in alleviating abiotic stresses, as well as the lsi1 mutant (information about this mutant appears only in the Discussion part). The Authors should consult recent reviews on silicon action on plants and the mechanisms, for example
Khan, M. I. R., Ashfaque, F., Chhillar, H., Irfan, M., & Khan, N. A. The intricacy of silicon, plant growth regulators and other signaling molecules for abiotic stress tolerance: An entrancing crosstalk between stress alleviators. Plant Physiology and Biochemistry 162 (2021) 36–47
Etesami, H., & Jeong, B. R Silicon (Si): Review and future prospects on the action mechanisms in alleviating biotic and abiotic stresses in plants. Ecotoxicol. environmental safety 147 (2018) 881-896
Cooke, J., & Leishman, M. R. (2016). Consistent alleviation of abiotic stress with silicon addition: a meta‐analysis. Functional Ecology, 30(8), 1340-1357
The purpose of the research is not clearly presented. There is no main hypothesis to be examined.
Results - statistics in figures and tables seems to be separate for WT and for the mutant which does not allow correct comparison. It is not clear how the applied stress affected Si accumulation in the WT rice. The paragraph 2.6. Chloroplast ultrastructure (lines 166-180) has substantial mistakes concerning the basic terminology, such as “the internal basal cycts” (did the authors mean thylakoid mwmbranes?), vacuoles in the chloroplasts, bubbles in the matrix, etc. Fig 4 is not of good quality and the magnification in 4A seems to be different from the others. “Vacuoles” – to me they look like starch granules. Please reexamine carefully these results and describe them correctly.
It looks strange that the photosynthesis seems not affected by the stress but chloroplast ultrastructure changes – how the authors could explain this contradiction?
The results about ROS enzymes are inconsistent with the general expectation of Si mitigating effect on oxidative stress.
Discussion is not based on the possible mechanisms underlying the beneficial Si effect in stressed WT rice. Conclusion – line 420 – “enhancing the activity of antioxidant enzyme system” – actually only POX reacts as expected.
Minor remarks
lines 15-16 – “very few studies have been investigated the effects …” – have investigated
line 17 – “Si-detective mutant” – defective? Better definition of the mutant is needed
Lines 44-45 – incorrectly cited ref 5. In the text – “… form toxic products, such as malondialdehyde (MDA), which can damage cell membranes through solutes leakage” – not true, solute leakage is the consequence and not the cause of membrane damage. In ref 5 (Scandalios, J.G. Oxygen stress and superoxide dismutases. Plant Physiol. 1993, 101, 7–12) the following statement exists: "Peroxidation damage of the plasmalemma leads to leakage of cellular contents, rapid desiccation, and cell death"
Lines 52-53 –… factors….regulates
Line 225 – aerenchyma instead of aeration
Line 236 – Fig 1 should be fig 3
Line 339 – “every three seedlings” - unclear
Round 2
Reviewer 1 Report
I am happy with the author's reply. the manuscript is improved now and can be accepted min its current format.
Author Response
Dear reviewer:
Thank you very much for your valuable comments and suggestions on my manuscript. This manuscript has improved the introduction, results and discussion.
Finally, thank you again for your comments and suggestions. Best wishes to you.

Reviewer 3 Report
The revised version is improved but still far away from being publishable. The good things – statistics now allows comparison of all variants, the explanations on chloroplast ultrastructure are more correct, the purpose of the study is more clearly presented in the introduction. However, Discussion is still limited to Si effect observed in various stresses, and not about the possible mechanisms of Si stress mitigating action. Still unresolved issues:
Lines 17, 247, 352 – Si-detective mutant
Detective is a person, especially a police officer, whose occupation is to investigate and solve crimes. Defective – adjective - imperfect, damaged, injured (the meaning seen by the authors) Better – Si uptake and transport deficient mutant
Lines 44-45 – “which can damage cell membranes through solutes leakage and eventually 45 form toxic products, such as malondialdehyde (MDA) [5].” – again wrong meaning of this citation
Line 46 – “plants possess complex antioxidant defensive system” – but only three enzymes were studied here – SOD, unspecific POD and CAT – why?
Line 60 – “quiescence behavior confers due to” – totally uncler
Line 69 – sentence begins with .And
Line 82 – “but no effects for Si-defective mutant material” – but will have no effects… “Material” is needless
Results 2.1.Plant biomass – line 91 – “increased growth of WT” – actually in stem and leaves but not roots; line 92 – “biomass by…%” in non-stressed plants – not clear
Line 107 – “Under submergence stress, total root length… increased” – actually not significantly
line 181 – chloroplasts… were fusiform
lines 183-185 – “significant changes in the ultrastructure of rice leaves were presented under submergence treatment (Figure 4B and E). The cell matrix was filled with a large number of starch grains, and plasmolysis occurred” – leaves, cell matrix, plasmolysis – not seen in this figure . Starch granules should be in the chloroplasts. Plasmolysis should be indicated.
Discussion
Line 231 - 3.1. – “Si application increased Si uptake and improved plant growth under submergence stress” – only in WT rice. Ambiguity - is Si uptake enhanced under stress, or unchanged, or diminished in WT rice plants?
Line 234 – “and many other traits” – unclear, too general
Line 235 – “both rice materials” - both WT and mutant genotypes
Line 292 – “the window hypothesis “ - comment is needed - this study confirms or not this hypothesis
Line 297 – improved or stabilized?
Line 303 – “the number of…” - morphometric data are missing, only one picture is shown
Line 317 – “enhanced antioxidant response under submergence” – data are not convincing for such a statement
POD - Non specific Class III plant peroxidases have diverse functions in plants such as in cell wall metabolism, lignification, suberization, ROS metabolism, wound healing, etc
Line 335 - CAT – “the activity of CAT in lsi1 mutant was not significantly reduced, possibly due to the consumption of large amounts of non-structural carbohydrates” – unclear. Catalase scavenges H2O2 generated during mitochondrial electron transport, β-oxidation of the fatty acids, and most importantly photorespiratory oxidation. Catalase is about 10-25% of the total peroxisomal protein.
At the end of the Discussion a conclusion is missing as a logical final - how Si helps rice plants to mitigate submergence stress, Maybe to put here the Conclusion and not after MMs
Line 380 – “scan using scanner (Epson” - scan (using scanner Epson
Line 439 – “In addition, Si reduced oxidase damage by enhancing the activity of POD and CAT…” oxidative damage… enhancing POD, maintaining CAT
